# Shared Micromobility: Between Physical and Digital Reality

**Daria Bylieva** [1,*], **Victoria Lobatyuk** [1] **and Irina Shestakova** [2]

1   Social Science Department, Peter the Great St. Petersburg Polytechnic University (SPbPU),
    195251 Saint-Petersburg, Russia; lobatyuk_vv@spbstu.ru
2   Department of Philosophy, Saint Petersburg Mining University, 199106 Saint-Petersburg, Russia;
    irina_shestakova@inbox.ru
*   Correspondence: bylieva_ds@spbstu.ru

**Abstract:** Moving around the city is a problem for the development of most megacities. Due to digital technologies, each city dweller is connected by information and communication channels with the city infrastructure, receiving information and choosing the available modes of movement. Shared micromobility in terms of digital solutions is a convenient service, while reducing congestion and emissions, and preventing air and noise pollution; however, the physical and social dimension of the city is experiencing problems, with growing public health concerns, high overall environmental costs, clutter in the streets, etc. This presentation presents a case study of the relatively recent emergence of shared micromobility in St. Petersburg and attitudes towards its users. In addition to the direct process of use and the experience gained, the factors that determine the social influence and perception of micromobility are highlighted. The highest ratings of the digital component and the rather high importance of such factors as environmental friendliness and safety make it possible to recommend the creation of an interactive digital system that unites riders.

**Keywords:** shared micromobility; digital reality; city; e-scooter; sharing





## 1. Introduction

Digitalization is changing human existence in many ways. Work, study and entertainment are increasingly influenced by information and communication technologies [1–5]. The COVID-19 pandemic has increased the pace of the digitalization of both public and private life [6–8]. New technological solutions are attractive due to the acceleration of processes, the automation of many functions, cost efficiency, etc.; however, while some processes can be almost completely transferred to a digital environment, others contain processes of the physical world as a necessary component. In the latter case, digital solutions have a particularly strong impact on the human environment and have a variety of social, economic, environmental and/or other consequences.

Moving around a metropolis is a serious urban development problem. The inability of the transport infrastructure to cope with increased traffic flow and passenger loads leads to the search for a variety of digital solutions. Digital systems called Intelligent Transportation Systems, Advanced Management Systems and Smart Mobility Systems allow data collection using video cameras, radio-frequency identification, global positioning system (GPS) data, media access control, and analysis using predefined, micro-simulated or agent-based models [9], and they optimize the elements of the transport structure of large cities. Digital technologies allow, in particular, smart parking [10], advanced traffic light systems [11], smart highways [12], vehicular ad-hoc networks [13], mobility as a service [14], etc.

Nevertheless, if most transport decisions can only be made at a centralized level, then single-track vehicle shares can appear quite 'spontaneously'. Like other sharing platforms, shared micromobility means a digital solution that replaces property with sharing, something which is in line with the current but critical trend toward an efficient

use of resources. Sharing bicycles and scooters has spread or gained a new lease of life with digital technology, making single-track vehicles fast and easy to use. Thus, it is digital solutions that make shared micromobility attractive to urban dwellers.

Understanding how and why micromobility services are used and how citizens value them is essential to building efficient micromobility ecosystems that contribute to a more sustainable urban transportation system. Hosseinzadeh, Algomaiah, Kluger, and Li consider micromobility as part of the Smart City paradigm [15] which, according to many authors, is the key to enhancing sustainable urban development [16]. Information and communication technology is becoming an essential element for accelerating progress on achieving sustainability [17]. Sustainable transportation in cities should strive to optimize its environmental impact (e.g., reducing greenhouse gas emissions, distance, fuel consumption, pollution, empty miles whilst increasing vehicle load) and social impact (e.g., increasing accessibility, reliability, health and safety whilst reducing congestion) [18]. This study offers a multilevel consideration of the cyber-physical-social reality. Studying this phenomenon, taking into account the digital, social and physical dimensions, it reveals the weaknesses and potential in the development of two-wheeled vehicles as a way of contributing to sustainable urban development.

The model of shared micromobility proposed in the article, combining the digital and physical reality, allows us to see both the problems that need to be solved in the physical and social environment, and possible solutions at the intersection of the social and digital dimensions. The aim of the study is to identify, on the basis of a survey of micro-transport users, the potential for increasing the stability of shared micromobility as a complex cyber-physical-social system in the urban space.

### 1.1. Combining the Digital, Social and Physical Space through the Example of Shared Micromobility

Due to digital technologies, every citizen is connected by information and communication channels with the city infrastructure. Today, building their own city routes, people using a mobile phone evaluate their options based on an online map showing the location of passenger transport and the presence of traffic jams, and they have the opportunity to use car-sharing, point-to-point cars or shared micromobility services.

The role of the bike and electric scooter sharing system in the infrastructure is an urgent problem of the modern city. The increase in the number of people traveling in two-wheeled vehicles changes the dynamics of movement and requires special regulation depending on the specifics of the urban infrastructure. Although bike-sharing systems can be traced back to 1965, it was digital technology that made them convenient and affordable. Currently, there are a number of options for shared micromobility: station-based and dockless bike sharing, electric bike sharing and electric scooter sharing [19]. Micromobility as a new and significant phenomenon of urban life in recent years has attracted the attention of researchers around the world, specifically in Greece [20], the USA [21], Italy [22], Germany [23], Denmark [24], Thailand [12], China [25], Austria [26], France [27], Australia [28], New Zealand [29], Saudi Arabia [30], and Singapore [31].

The application for a mobile phone allows you to find the nearest place for a bicycle or scooter, use and leave the transport in a convenient place and pay for travel. In addition to ease of use, researchers note their economic [32] and environmental [33,34], efficiency, a reduction in congestion and emissions, and the prevention of air and noise pollution. From the point of view of urban management, such services not only help to bridge the 'last mile' gap but as a rule do not require investment. On the contrary, they pay fees and also facilitate easier access to urban centers where parking is scarce and motor vehicle travel is more difficult [35].

Considering the relationship between the user and the digital environment, shared micromobility represents the ideal of public transport, when transport as an object of the physical environment is ideally integrated into the digital-physical system (Figure 1) of round-the-clock satisfaction of the urgent need of citizens for movement. The digital

environment of micromobility represents a reflection of the key factors of the physical environment, primarily demonstrating the location of the nearest vehicles and possible parking spots. In addition, the digital environment permits an optimization of the route by means of maps which reflect the situation on the road. This said, many important factors of the physical environment today are still not reflected in cyber reality.

**Figure 1.** Cyber-physical dimension of shared micromobility.

The social consequences of using single-track vehicles are currently not optimal. Dockless transport clutters the street and interferes with the passage of pedestrians. According to a study in Rosslyn, Virginia, 16% of e-scooters were not well parked while 6% blocked pedestrians [36]. Where single-track vehicles should move—on footpaths or only on roads and bike paths, especially when these are scarce—is a serious road safety problem. Many cities have adopted ad-hoc policies, sometimes imposing substantial fines on operators or users not complying with traffic rules [37]. There are various ways in which local authorities regulate where and how fast an electric scooter can travel. Thus, in different US jurisdictions, the policy of riding e-scooters on sidewalks is either permitted or prohibited, or permitted only outside of the central business districts, with the maximum permitted speed being from 10 to 20 mph [38]. E-scooters, despite their rapidly-growing distribution process, are fully regulated in Germany, France, Singapore, New Zealand and Australia. In other countries, the regulatory process has either just begun, or this new mode of transport has not yet attracted the attention of legislators. The lack of infrastructure necessary for movement in many cities is becoming a serious obstacle to the spread of micromobility [39].

The biggest problems are caused by e-scooters, which are becoming an increasingly popular form of transport due to their significant travel speed; however, the same speed makes this form of vehicle especially dangerous for both the rider and others, leading to injuries—something which has recently caused growing public health concerns [40,41]. Motor vehicles, other scooters and bicycles, infrastructure, and animals can also be injured [38]. There are precedents where e-scooter policy has undergone fundamental changes following

the death of an e-scooter rider in an accident, as happened in Tulsa, Oklahoma, and Elizabeth City, New Jersey [42]. Research shows that even using an e-bike instead of a regular bike makes riders' behavior more dangerous to those around them [43]. While the public health benefits of cycling are clear [44], there is no definite answer regarding the electronic scooter. On the one hand, researchers see them as a threat to healthier forms of movement such as walking and cycling [45], and on the other hand, moving while standing in the air seems healthier than being in a bus or car.

Sometimes the short service life of shared transport means not only the transformation of city streets, and sometimes parks and even water bodies into a scrapyard, but also high environmental costs [46,47]. The need to collect and move (and in the case of an e-scooter, also recharge) shared transport without a docking station means additional environmental costs that are sometimes ignored when describing this 'green' transport. Where bicycles are left and the logistics policies of companies are also proving to be environmentally important factors. In this case, it is possible to create a user-based relocation scheme based on pricing benefits for the user [48].

The attitude to shared micromobility services by the population, authorities and users in different cities varies significantly. Often, decisions to regulate sharing are made at the local government level after problems with free-floating bicycles and scooters have arisen, demonstrating that the lightness and simplicity of digital solutions must be combined with thoughtfulness at the level of the physical urban space in which both parked and moving vehicles should be accommodated.

*1.2. The Social Dimension of Shared Micromobility*

Much attention has been paid to the various aspects of shared micromobility in recent years. Nevertheless, this phenomenon remains unknown in its entirety as perceived by its users. Shared micromobility as a new and unique phenomenon turns out to be not just a way of moving in the city, but is associated with social ideas and influence, perceived simplicity, usefulness, safety of use, personal experience, safety awareness and the environmental friendliness of this type of vehicle.

Juelin, Lixian and Junjie found that the practice of riding plays a central role in the value co-creation for shared micromobility [49]. At the center of the social dimension scheme, shared micromobility (Figure 2) is the riding of micro-vehicles. Here a distinction can be made between the degree of emotional satisfaction from the riding process and personal achievements directly related to the experience. The researchers note that the use of micro-transport is characterized by affective values, i.e., an element of playfulness that appears to have considerable appeal [37]. In addition, one can single out the components of assessing one's own experience: emotional, associated with subjective well-being [50], and rational, associated with the perceived usefulness of using the service, as well as trust [51] as a cumulative assessment of experience; however, many other social constructs will influence these factors.

Within the framework of social learning theory, it is assumed that people are influenced by another person's behavior, and they adopt the behavior themselves if they evaluate its results as positive and what is socially approved [52]. Chen and Chao associate social performance with subjective rationalization [53]. Shen, Sun, and Wang write about the social value of using an object that is socially approved [54]. Moreover, the social environment can have several vectors of influence. The influence of public opinion on the use of micro-vehicles is two-way. Public opinion influences the decisions made about the use of scooters and bicycles, but the use itself influences public opinion as well. In addition, a positive experience of use can have a direct impact on the immediate social environment if a person shares their positive experience and offers to participate in an exchange. Here it is important to highlight the influence of the social environment, and especially the reference group of people, whose ideas have a vector of focus on a person.

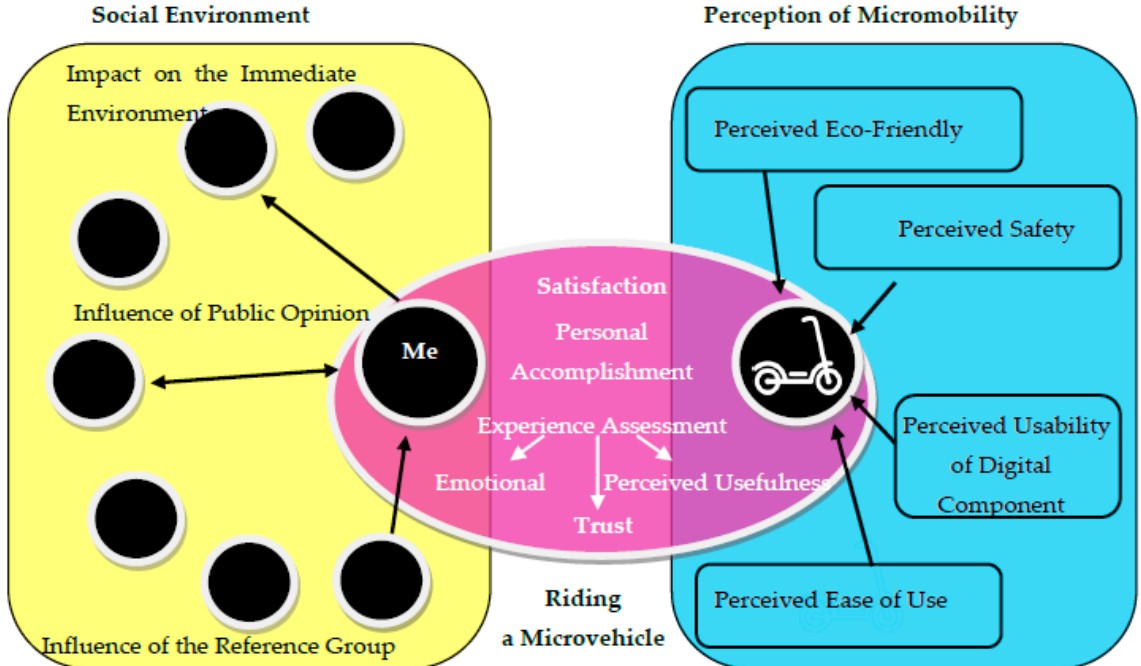

**Figure 2.** Key factors in the social dimension of shared micromobility.

The next group of factors relates to the perception of the transport itself. What constitutes its 'image' in the eyes of users? Here we have highlighted the perceived ease of use. Perceived ease of use is one of the main factors responsible for a willingness to use a technological innovation. The Technology Acceptance Model describes how users come to accept and use technology through its perceived usefulness and perceived ease of use [55]. The remaining three elements are factors that make up shared micromobility as an element of the cyber-physical reality of the city: the perceived usability of the digital component, the perceived security and the perceived environmental friendliness. Studies show that safety in many cases topped the list of reasons for not using e-scooters [42,56]. Caring for the environment has also recently become an important determinant of behavior and green perceptions impact the adoption of sustainable innovations [57,58].

## 2. Materials and Methods

In 2021, a survey of residents of St. Petersburg (*n* = 1500) was conducted amongst the population (N = 5,384,342). The survey was conducted only among adults aged 18 and over. Additionally, respondents who had no experience renting scooters and bicycles were immediately identified and eliminated from the survey, Answers from 285 experienced user participants follow. Table 1 presents more detailed demographic characteristics of the respondents.

**Table 1.** Demographic profiles.

| Profiles | Description | Percentage (%) |
|---|---|---|
| Gender (Male/Female) | Male | 60.1 |
| | Female | 39.9 |
| Age | 18–20 | 83.4 |
| | 21–25 | 10.6 |
| | 26–30 | 1.4 |
| | 30–40 | 1.1 |
| | more than 40 | 3.5 |

The majority of respondents identified themselves as students (87.7%), and the remaining 12.3% were evening or distance- learning students, that indicated in the survey their work status. Additionally, in the presentation of data regarding the respondents, information on the vehicles they used has been indicated, Therefore, only 7.7% had used only bicycle sharing, 83.5% had used only scooters, and 8.8% had used scooter and bicycle sharing.

In 2021, the shared micromobility in St. Petersburg was in the early stage of development and it was used only by a small proportion of the population. Due to a greater propensity for innovation, mobility, and certain image features, young people are the basic audience for scooter sharing. According to the Whoosh service, which is one of the largest in Russia and St. Petersburg, the main shared micromobility user is a young city dweller, and most often a student [59].

We have used an interdisciplinary approach that allowed us to view the micromobility sharing of micro-mobile devices as an example of the integration of digital, social and physical space in terms of the theories of sustainable development, urbanism and sociology. In addition, scientific research methods such as the construction of logical diagrams, and the graphical interpretation of theoretical information and empirical data were used. One of the main research methods was a sociological survey of the residents of St. Petersburg. The use of the latter made it possible to collect the main block of information about the research topic. In addition, statistical and mathematical methods were also used to process the questions and answers of the respondents.

It should be noted that the main part of the survey was made up of responses with a 10-point Likert-type response format with values ranging from strongly agree (10) to strongly disagree (1).

In this study, with the help of Alf Cronbach, we checked the questionnaire. The total value for all 3 blocks(factors) of questions exceeded 0.98, which indicates a high internal coherence The collected data were analyzed using output statistics (Pearson correlation coefficient for calculating the correlation value between variables) using the statistical analysis program SPSS 20.

The respondents took part in the research on a voluntary basis. The study results were anonymized with regard to names and to any other links that may identify the individuals. Ethical approval was received from the Ethics Commission founded by the Institute of Humanities, Peter the Great St. Petersburg Polytechnic University, which is ruled by the code of ethics of the Russian Society of Sociologists.

## 3. Results

### 3.1. The History of Sharing Bicycles and Scooters in St. Petersburg

St. Petersburg is the second largest city in Russia with a population of 5.4 million people and a total area of 1439 km$^2$. The historical center of the city and related complexes of monuments are included in the UNESCO World Heritage List. Seventy percent of the townspeople travel by public transport. The average travel distance to work is about 16 km, the travel time is 64 min on the road network and 54 min on urban passenger transport. Around 6.55 million emails are sent per day. Among the main problems identified in the framework of the concept for the development of the transport system of St. Petersburg are transport congestion of public transport during peak hours, a suboptimal route network (density 7.9 km/km$^2$), unresolved problems connected to the parking of private vehicles and the insecurity of the transport complex [60]. The length of city streets is about 3.5 thousand km. The length of bike paths is about 100 km.

The world history of e-scooter exchange started in 2017 and bike-sharing has been around since 1965. For St. Petersburg, however, the history of micromobility began in 2014 with one bike-sharing service with 250 bicycles and 29 bike stations in the city center. In 2017, there were a maximum of 96 stations, in 2018 there were 56 stations, and in 2019 there were 20 stations left when the city administration, which initiated its appearance, decided to withdraw support. In 2020, this support was replaced by a new city firm using bicycles

without stations. Instead, it was supposed to use the existing public bicycle parking lots in the city (Figure 3), of which there are about 1300; however, not all parking-lot owners agreed with this format of work, and the company was forced to install new primitive bicycle-parking lots, including at the site of the dismantled parking lots of its predecessor. Consequently, cyclists sometimes leave their vehicles outside parking lots. By 2021, the company already had over 2000 vehicles and in addition to this sole bike rental company, there are also several bike rental points in the city that allow members of the public to take a bike for time-periods ranging from several hours to several days, but the number of such rental points is low.

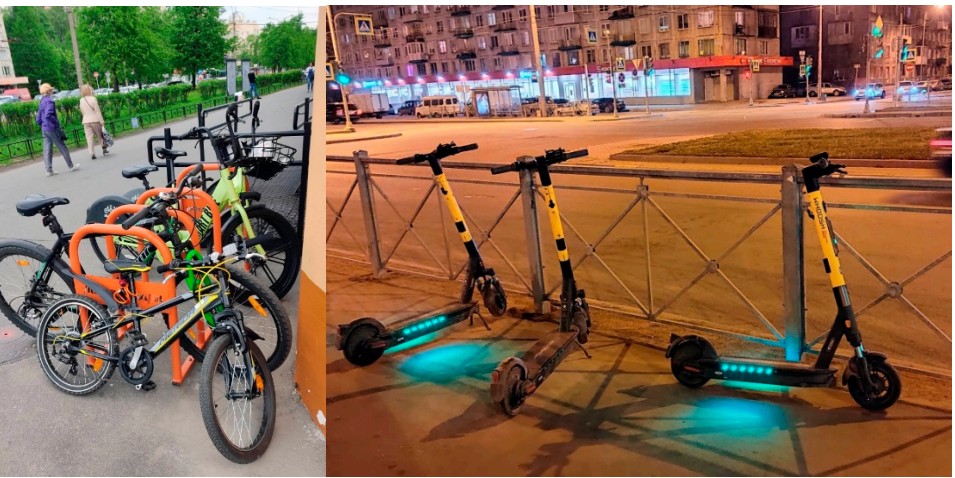

**Figure 3.** Store-owned parking for bicycles with private and shared bicycles. Free floating e-scooter. St. Petersburg, 2021.

The first sharing scheme appeared in St. Petersburg in October 2018 with 48 scooters and 6 stations, but without the consent of the city administration, thus, it was declared illegal. In 2019, a new service with 50 electric scooters was opened in the city. When the scooters were attached to city fences instead of stations, the number of rental scooters began to skyrocket and by the end of 2019, there were about 4000 electric scooters in promotional rings, and about 8000 in 2020. In 2020, more than 10 e-scooter sharing companies operated in St. Petersburg, but attempts by companies to agree on the location of stationary parking have not been crowned with success; of the 300 addresses from whom permission was sought, only about 3% agreed. The reasons for refusal in the most popular and busy locations of the city were the narrowness of the streets, and the presence of monuments, old buildings, and underground structures. As a result, the fastening of scooters to fences has been replaced by free-floating models with virtual parking lots. In 2021, the total fleet of electric scooters in the city amounted to 12,500 scooters, of which 11,000 belonged to the three largest market players.

In early June 2020, an initiative was launched to create a single digital map with full interactive information on all the shared micromobility, but the project is still in the planning stage.

From the beginning of 2021 to August, the number of accidents with e-scooters in St. Petersburg was 32 [61]. The e-scooter does not have its own legal status in Russia and is not included in the traffic rules as a special object, something which has made it difficult to determine the legal consequences for riders who have committed traffic violations. Consequently, for example, according to Russian legislation, a bicycle rider can be required to pay 1000–1500 rubles for causing mild and moderate harm to the health of an injured person, and a moped (a vehicle with an engine power of more than 250 W) driver can be required to pay from 2500–20,000 rubles, with the possibility of a loss of driving rights for 1.5–2 years, depending on the severity of the injuries [62,63]. Whereas a person using roller skates, scooters or other similar means for movement is considered to not be a rider,

but a pedestrian. If therefore, when reporting an accident on a sidewalk, the police do not register the scooter as a vehicle (in most cases they do not), its rider cannot be made administratively or criminally liable. Victims can only recover compensation through a court, which considers the matter to be a dispute between two pedestrians.

In June 2021, after several accidents, including those involving injured children, the city administration drew attention to the problems associated with free floating electronic transport. As part of the investigation of cases under the article, *Provision of Services That Do Not Meet Security Requirements*, searches were carried out in the five largest e-scooter-sharing services in St. Petersburg. On 9 June 2021, scooters completely disappeared from the streets of the city and mobile applications stopped working. As it turned out, the companies had previously signed an agreement with the city administration regarding the rules of their work and by the evening of the same day, an agreement was adopted on limiting the speed of the e-scooter, namely, that the speed in the city should not exceed 20 km/h, and on the sidewalks, 15 km/h. A special regulation came into effect at the time of the football championship in June–July 2021, when riding on 95 central streets was banned and the speed on sidewalks was limited to 10 km/h. The large number of scooters, parked and sometimes just scattered around the city, caused a public outcry. An anti-theft alarm was trigged by those who tried to move a vehicle causing an obstruction on their own initiative. Parking restrictions were also introduced and it was no longer legal to park on sidewalks less than 1.5 m wide; closer than 15 m from transport stops or metro stations; in parks, or in green spaces, etc. Companies were ordered to provide scooters with unique numbers, and there was a ban on their use by persons under 18 years of age and by the intoxicated [64].

However, the measures taken are not final and do not fully satisfy the various stakeholders. At the time of writing, the possibility of introducing electric scooters into the legal field and of building a digital urban ecosystem are being discussed.

### 3.2. Shared Micromobility in the Minds of Petersburgers

The empirical part of the study is presented in the form of a survey of residents of St. Petersburg. In the graphs below, information about the first call, the frequency of calls, the average usage time per day, as well as the average number of kilometers traveled using these services can be found. We will take a closer look at scooter users, since this form of transport significantly predominates in terms of the number of users and vehicles in the city under consideration.

Figure 4 shows the frequency of use of scooters with 52% of respondents having indicated the answer 'from one to several times a month', 22% of respondents having used the service 'from one to several times a week' and 26% of them having used it 'from one to several times a year'. This indicates that the majority of respondents quite often used micromobility services in their daily life, and that for them it is a permanent mode of transport.

Figure 5 shows the answers of the respondents regarding when they began to use these services. The answer 'this year' was the most popular among the scooter users (62%), while 31% indicated 'more than a year ago', and only 7% of respondents had used micro-mobile sharing for 'more than two years'.

Figure 6 provides information on the average sharing time per day with 60% of respondents having used a scooter for 'less than an hour', 36% having used a scooter for 'one to three hours', while only 3% indicated long-term use from 'three to six hours'. Only 1% had used a scooter for 'more than six h'.

Figure 7 shows the average distance traveled by users in shared vehicles, with most of the respondents (50%) indicating the option from 'two to five kilometers'. In second place was the answer 'up to two kilometers', which was recorded by 30% of scooter users. 'From five to ten kilometers' was chosen by 16% of the respondents, and 4% chose long routes of more than ten kilometers.

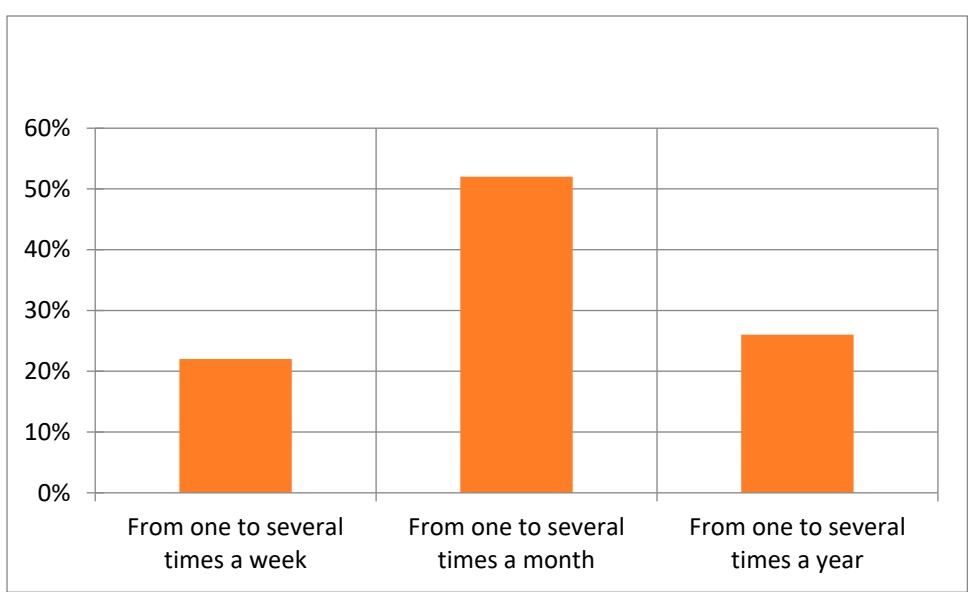

**Figure 4.** Frequency of use of the scooter-sharing.

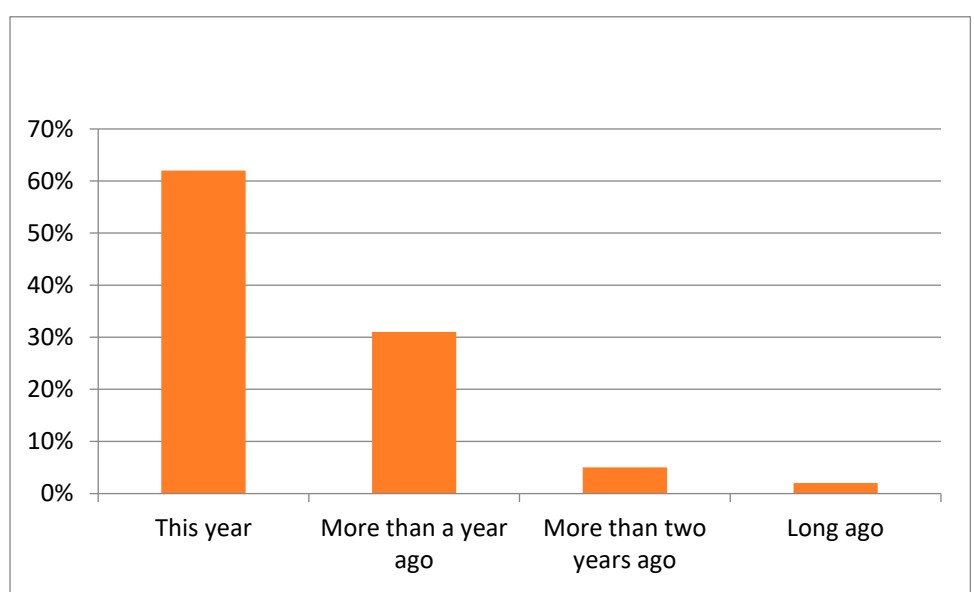

**Figure 5.** Start of using the sharing service.

Users of such bicycle services generally do not differ much in the answers presented in this study; they also preferred distances of up to five km (79% of respondents) and had turned to micro-mobile sharing services during this or the last year (83%). If we talk about the frequency of trips, then it was distributed evenly between the three options of answers: 'one or more times a week'–32%, 'one or several times a month'–36%, or 'one or more times a year'–32%.

The survey of residents of St. Petersburg in the form of a questionnaire consisted of 38 questions for cyclists and scooter users. The received answers were divided into three blocks of judgments, united by a common problem: riding a shared micro-vehicle, social environment and the perception of shared micromobility (Figure 2). Moreover, each block was divided into between three to five sub-blocks. Appendix A includes the statements for these blocks, as well as the results obtained from respondents in the form of averages.

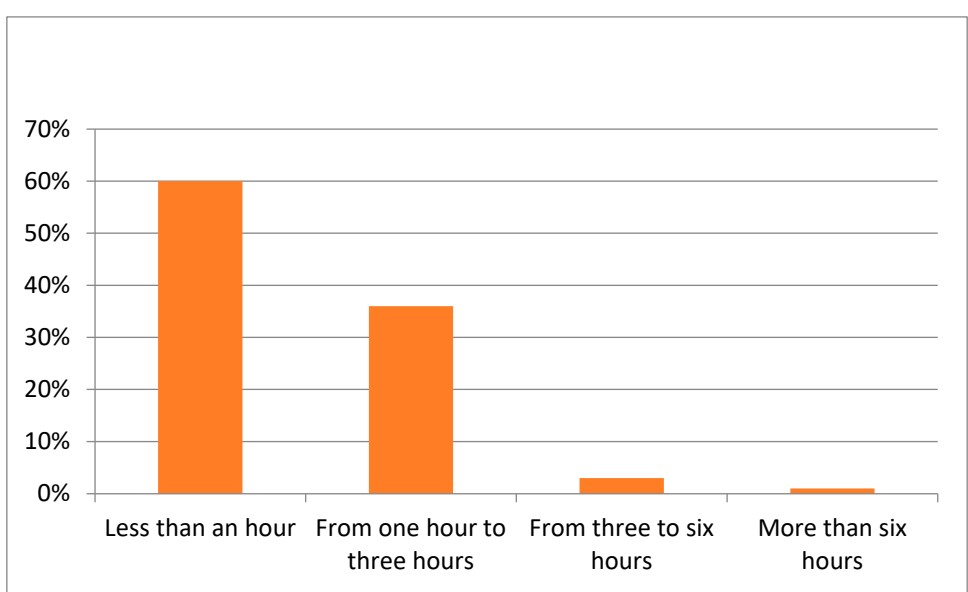

**Figure 6.** Sharing time per day.

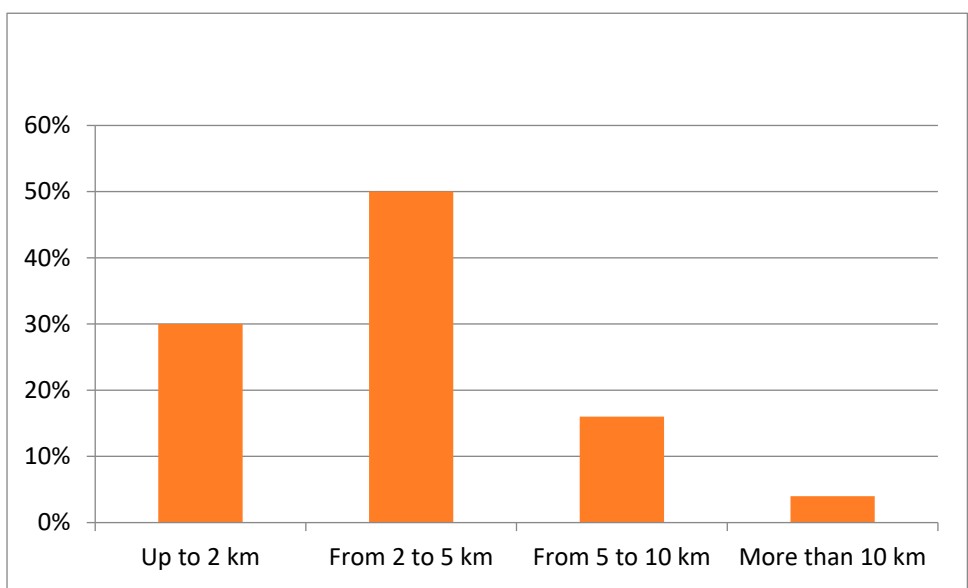

**Figure 7.** Distance the user moves using sharing.

Figure 8 presents a block of judgments about the process of using micro-mobile sharing, namely, riding a micro-vehicle. It included sub-blocks concerning the satisfaction with user experience, trust attitude, personal accomplishment, perceived usefulness and subjective well-being. The sub-block, satisfaction with user experience, had the greatest value for respondents, and this was noted by both the scooter and bicycle users. Users pointed out that sharing is enjoyable (7.4/7.3) and fun (7.3/7.2) and these were the most popular answers. The least popular response was that scooter-sharing/bike-sharing is exciting (7.0/6.8). The judgments about the benefits of scooter-sharing/bike rental had the average values of 7.2 and 7.3, respectively. Among users, the lowest ratings among the sub-blocks were personal accomplishment (6.43) and subjective well-being (6.6). Personal accomplishment was dedicated to assessing the importance of the impact of calories burned and kilometers traveled on the user. The respondents were more positive about the information regarding the miles traveled by the scooter per trip, and the average value for this issue was 6.2. The calories that users burned while traveling were not that important (5.9). This was probably due to the fact that the scooter is perceived as a

means of transportation and not as an opportunity to combine exercise with commuting. The situation is different with a bicycle, which according to these judgments shows the same values—6.8, therefore it can be assumed that the users often perceived this kind of movement as a combination of travel and exercise. The lowest mark in the subjective wellbeing sub-block was given to the statement 'I feel that after using the scooter-sharing my life has become more complete' (5,9).

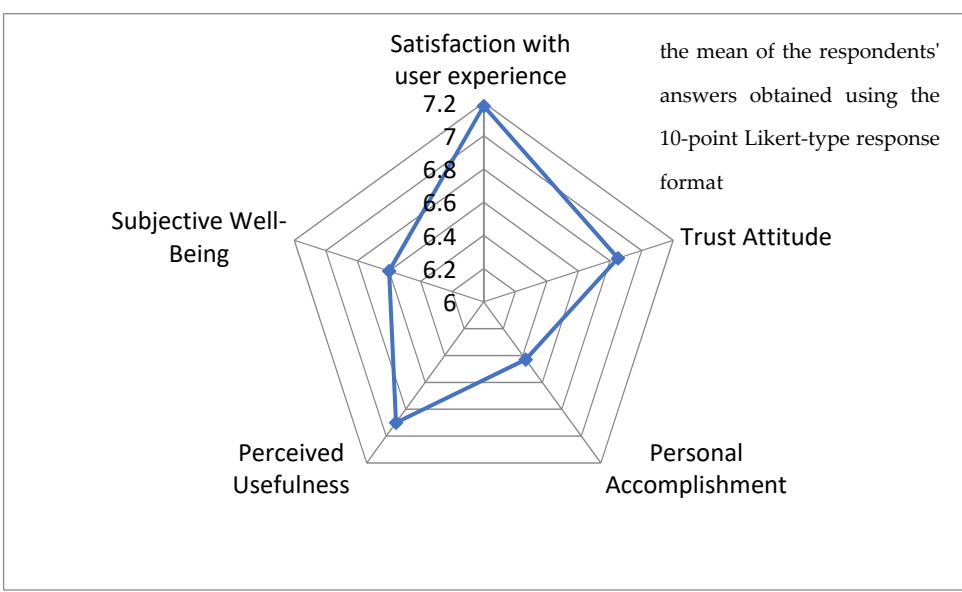

**Figure 8.** Factors for assessing micro-vehicle experience.

The following statements were also included in the sub-block: 'My experience of using a scooter/bike sharing was memorable and it was able to make my life easier' (6.6/6.7), and 'In general, I felt good shortly after riding the scooter' (6.7/7.1). These received higher than average values. The sub-blocks for trust attitude and perceived usefulness had an average of 6.85 and 6.9, respectively.

The statements, 'Based on my own experience of scooter/bike sharing, I know this system is not dangerous' (6.8/6.9), and 'Based on my own experiences with scooter/bike sharing, I know this is in line with user expectations' (6.9/6.8), in the trust attitude sub-block had almost the same meaning to respondents. The perceived usefulness sub-block showed a wider range of mean ratings among the survey statements. Thus, the highest ratings (7.0–7.1) were given for the convenience of travel scooter-sharing/bike-sharing, as well as their efficiency and practicality. The least significant was in the statement, 'I believe scooter sharing/bike sharing is beneficial in my daily movements'—6.6/6.5. On the basis of this, we can put forward a hypothesis that micro-mobile sharing is a convenient, but not an everyday mode of transport, and one which is only chosen in certain circumstances. Table 2 presents a summary of the statistics of factors for assessing the micro-vehicle experience.

**Table 2.** Summary of statistics of factors for assessing micro-vehicle experience.

| Average | Minimum | Maximum | Scope | Variance | Alf Cronbach | Number of Points |
|---------|---------|---------|-------|----------|--------------|------------------|
| 6.772 | 5.829 | 7.363 | 1.263 | 0.212 | 0.981 | 17 |

Figure 9 shows a diagram of the third block of judgments, revealing the perception of shared micromobility (sub-blocks: injury risk, environmental friendliness, convenience of the digital environment, perceived ease of use). In general, all judgments received high marks, as people appreciated the opportunities and advantages, but also recognized the

danger of this type of transport. At the same time, the judgments from the sub-blocks, convenience of the digital environment (7.03) and perceived ease of use (7.13), received the greatest support from the respondents. There was a lower value for the blocks associated with injury and environmental friendliness.

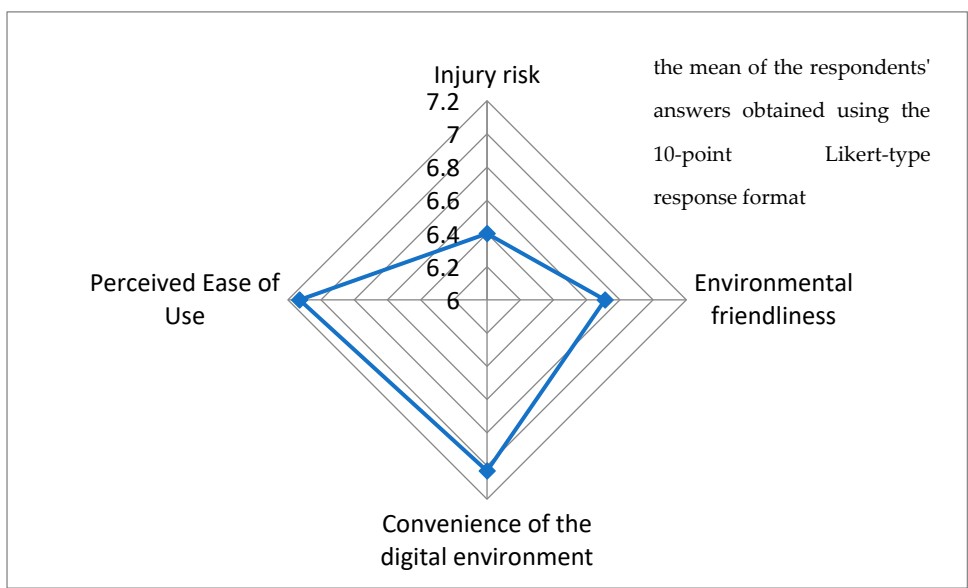

**Figure 9.** Factors in the perception of shared micromobility.

The statement, 'I use scooter sharing because the scooter/bike is an environmentally friendly mode of transport' had an average value of 6.5/6.8. The statement 'I advise my friends and family to use scooter sharing whenever possible; since the scooter does not harm the environment' scored 6.4 and 6.7, respectively, whilst the statement 'An inexperienced user of sharing can injure others' obtained a score of 6.0. Thus, the respondents noted that it was the lack of separate scooter lanes (and the small number of bike lanes), rather than a lack of experience in using these vehicles that posed a threat to those around them. The most popular judgment of the block was 'I like the fact that I see the nearest scooter sharing/bike sharing station in the mobile application' (7.5/7.3), something which undoubtedly indicates the further development of these services related not only to the expansion of the capabilities of mobile applications but also to the creation of a specialized infrastructure. Table 3 presents a summary of the statistics of factors in the perception of shared micromobility.

**Table 3.** Summary of statistics of factors in the perception of shared micromobility.

| Average | Minimum | Maximum | Scope | Variance | Alf Cronbach | Number of Points |
|---------|---------|---------|-------|----------|--------------|------------------|
| 7.000 | 6.402 | 7.504 | 1.102 | 0.131 | 0.958 | 12 |

Figure 10 shows a diagram of a block of judgments about the social environment, and it includes three sub-blocks: influence of public opinion, influence of the reference group, and impact on the immediate environment. The judgments from the sub-block, impact on the immediate environment (6.06), received the highest marks; the questions from this sub-block related to the sharing of information about micro-mobile sharing with family and friends, as well as advice on its use (the highest average scores were for 'I often recommended scooter-sharing/bike sharing to my family and friends (6.2/6.3)', and 'I often use scooter-sharing/bike-sharing to relax with my family and friends' (6.3/6.0)). Conversely, fewer respondents indicated that their acquaintances had begun to use such services on their advice (5.8). The smallest values were for the sub-blocks: 'Influence of

public opinion' (5.77), and 'Influence of the reference group' (5.71). The smallest value in the sub-block 'Influence of public opinion' was represented by the questions 'When I rent a scooter, I look better in the eyes of others' (5.2), and 'Using scooter-sharing gives me social approval' (5.4). The same statements among bicycle users scored higher values: for the first it was 6.0, and for the second, 6.3. In the sub-block 'Influence of the reference group', the statement questions that had the lowest scores were, 'Most people who are important to me think that I should use scooter-sharing', and 'Most people who are important to me think that using a scooter-sharing is a good idea' (5.5). Slightly more important for the respondents was the statement, 'Most of the people influencing my decisions think that I should use scooter/bike sharing' which resulted in scores of 5.7 and 6.0, respectively. Table 4 presents a summary of the statistics of factors in the perception of shared micromobility.

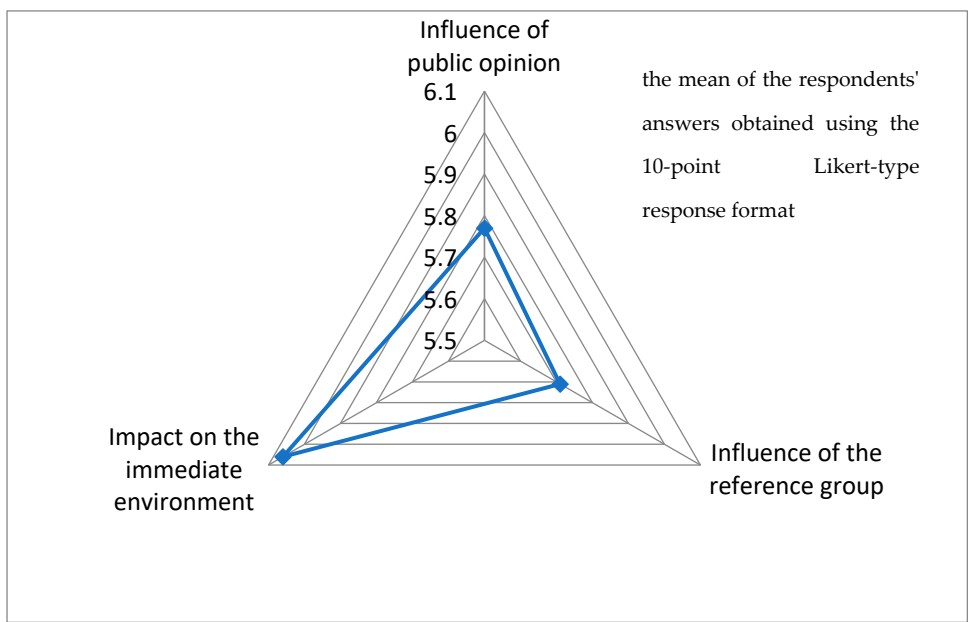

**Figure 10.** Factors assessing the impact of the social environment on the use of shared micromobility.

**Table 4.** Summary of statistics of factors in the perception of shared micromobility.

| Average | Minimum | Maximum | Scope | Variance | Alf Cronbach | Number of Points |
|---------|---------|---------|-------|----------|--------------|------------------|
| 5.785 | 5.271 | 6.341 | 1.071 | 0.158 | 0.963 | 9 |

The diagram in Figure 11, where a 10-point Likert-type response was also applied, shows the average values for the three blocks (Figure 2), with bicycles and scooters being considered together. Thus, we see that the social environment had the least influence on users (average value 5.84), while the perception of micromobility (average value 6.82) and the process of use itself (average value 6.79) had almost the same effect. If we consider only the scooter users, then the impact of social environment would be even less, and would be only 5.68, with the perception of micromobility being 6.86 and the process of use being 6.68.

Table 5 shows the correlation coefficients between the main blocks. We found high correlations on the Chaddock scale ($0.7 < r < 0.9$) between the factors, 'satisfaction with user experience', 'influence of public opinion', and 'influence of the reference group'; 'injury risk' with 'trust attitude', 'personal accomplishment', 'perceived usefulness', and 'subjective well-being'; and 'perceived ease of use' and 'convenience of the digital environment' with all factors assessing the impact of the social environment on the use of shared micromobility. The authors point to the presence of correlation coefficients of more than 0.9 between several factors; this can become the basis for finalizing the hypotheses at the next stages of the study.

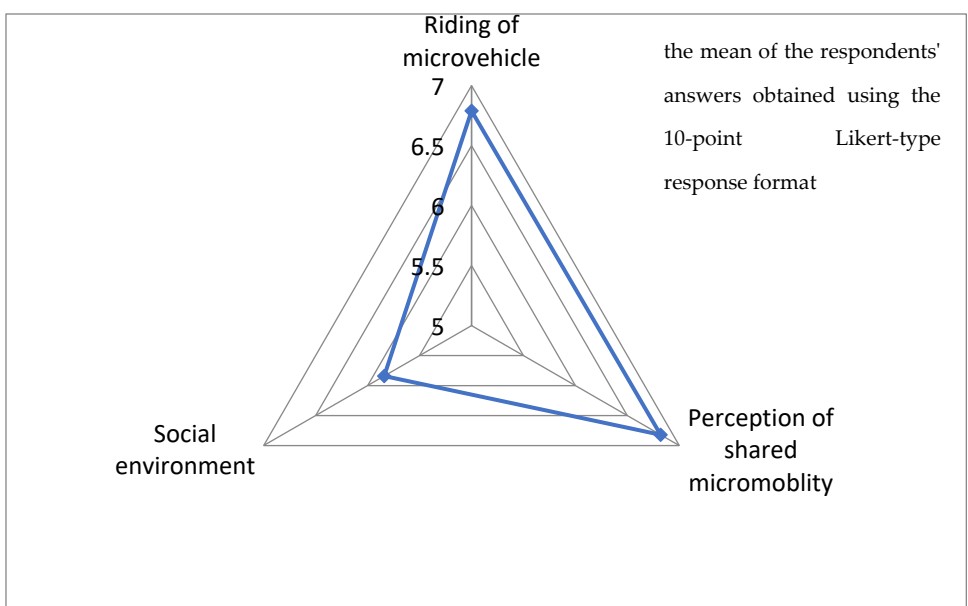

**Figure 11.** Average score of shared micromobility by blocks.

**Table 5.** Correlation coefficients among variables of the study.

| | Trust Atti-tude | Personal Accom-plish-ment | Perceived Useful-ness | Subjective Well-Being | Injury Risk | Environmental Friendliness | Convenience of the Digital En-vironment | Perceived Ease of Use | Influence of Public Opinion | Influence of the Refer-ence Group | Impact on the Im-mediate Environ-ment |
|---|---|---|---|---|---|---|---|---|---|---|---|
| Satisfaction with User Experi-ence | 0.969 ** | 0.890 ** | 0.977 ** | 0.953 ** | 0.906 ** | 0.954 ** | 0.972 ** | 0.969 ** | 0.878 ** | 0.898 ** | 0.915 ** |
| Trust Attitude | | 0.903 ** | 0.964 ** | 0.958 ** | 0.882 ** | 0.955 ** | 0.954 ** | 0.963 ** | 0.893 ** | 0.907 ** | 0.925 ** |
| Personal Accom-plishment | | | 0.896 ** | 0.919 ** | 0.811 ** | 0.905 ** | 0.875 ** | 0.865 ** | 0.903 ** | 0.908 ** | 0.906 ** |
| Perceived Useful-ness | | | | 0.961 ** | 0.886 ** | 0.959 ** | 0.962 ** | 0.959 ** | 0.897 ** | 0.915 ** | 0.926 ** |
| Subjective Well-Being | | | | | 0.869 ** | 0.955 ** | 0.932 ** | 0.932 ** | 0.922 ** | 0.941 ** | 0.951 ** |
| Injury Risk | | | | | | 0.867 ** | 0.908 ** | 0.891 ** | 0.801 ** | 0.830 ** | 0.835 ** |
| Environmental Friendli-ness | | | | | | | 0.937 ** | 0.935 ** | 0.922 ** | 0.936 ** | 0.944 ** |
| Convenience of the Digital Environ-ment | | | | | | | | 0.962 ** | 0.854 ** | 0.871 ** | 0.888 ** |
| Perceived Ease of Use | | | | | | | | | 0.861 ** | 0.880 ** | 0.898 ** |
| Influence of public opinion | | | | | | | | | | 0.945 ** | 0.915 ** |
| Influence of the reference group | | | | | | | | | | | 0.941 ** |

**. correlation is significant at the 0.01.

## 4. Discussion

It is necessary to establish special digital models that take into account not only the physical urban space and traffic dynamics, but also the social dimensions of micromobility

sharing. The digital model should be built not only taking into account the existing traffic flow but should also have a positive effect on changing preferences.

St. Petersburg, as one of the largest cities in Europe, suffers greatly from traffic jams and a lack of parking spaces. Thus, micromobility sharing can prove to be a valuable resource with a sustainable implementation in the cyber-physical space. Different cities differ not only in their congestion of roads, and the space allocated for cycling and walking, but also in the perception and way of using the new type of movement by its users. For example, bike-sharing systems have different impacts on traffic congestion in different cities, for example, larger cities become better off but richer cities become worse off [65]. These differences can be noted: (1) by objective indicators of use, for example, by the length of the run, which in St. Petersburg is longer than in the cities of Germany [66]; and (2) for the purpose of urban micromobility. For example, the most common destinations to which e-scooters are reportedly being ridden by riders of Provo, UT are 'just riding around for fun' (25.3%), home (20.0%), and dining/shopping locations (17.1%) [35]. In China, the use of bike sharing is based on physical exercise (51%), transfer to a bus or a subway (56%), as an alternative to walking, recreation and entertainment (36%), or going to work (26%) [50]. E-scooters are used for commuting in 20% of travel in France and may be a more frequent option in many cities, namely, Denver (more than 50%), Oslo (about 40%), Santa Monica, and Los Angeles (about 30%) [67]. In St. Petersburg, the main purpose of use is for recreation (75%), while the second rather popular option is as a way to get to a place of study or work (53% for e- scooters and 38% for bikes).

In cities where micro-exchange is a well-known reality, attitudes towards it have changed in a positive direction due to the need to keep socially distanced during the COVID-19 pandemic [68]. In St. Petersburg, micromobility sharing has become a novelty in recent years and it should be noted that the launch of micromobility during a pandemic placed it in a uniquely advantageous position. This said, we do not have the opportunity to compare its use with the period before isolation. Almost half of the riders surveyed began using them only in the current year, 2021, and a third in the previous year. The process of use was satisfying for users, but not for others and the assessment of the social environment was much lower. Here, both the ignorance of the townspeople about the novelties of the transport and the first negative precedents, which received a certain resonance thanks to social networks and local media, were reflected. How much the new transport will fit into the city depends largely on the few current users.

Examining the perception of micromobility sharing from a digital and physical perspective reveals the potential for expanding the digital space to address the challenges of the physical environment for greater sustainability. In particular, the results of this study confirm that digital technologies are most satisfying and appreciated by micro-vehicle users. Digital user immersion allows for the recommendation of the inclusion of aspects of the service that are important to riders in the cyber dimension of micromobility sharing. Riders' perception of micromobility sharing allows them to see development paths that meet sustainability goals and reflect a philosophy of lean sharing. The ecological component is also considered by them as an important aspect of the use of a shared micromobility. Thus, it is possible to integrate environmental performance and inter-user social interaction into the digital component of the service (Figure 1), in particular to highlight the parking problems associated with environmental costs and the ease of movement on the street. At the same time, in a digital environment it can, in addition, offer the possibility of creating an account with different ratings, competition and cooperation [69,70], gamification [71,72], and visual symbols [73,74]. The interaction of users in the digital environment to achieve a greater environmental efficiency of sharing can foster the feeling of belonging and reciprocity [75,76], which, according to Celata, Hendrickson, and Sanna, is characteristic of highly-connected sharing [77].

Problems related to the dangers that electric scooters can pose to the riders and those around them were somewhat less concerning for the respondents; however, having a

certain level of concern allows us to look for ways of multilateral cooperation, including not only firms and administrations, but also riders, to create a safe urban space.

This study is limited to a specific city case at a fairly early stage in the use of shared micromobility. Sufficiently high scores for all parameters can be caused by the fact that the respondents are early users, lovers of new products, and were ready for their positive perception. Therefore, it is necessary to conduct research after a certain period of time in order to determine the changing perception of micromobility in St. Petersburg. In addition to the factors considered, there are other factors affecting the use of shared micromobility services, in particular the weather conditions [78], which are also often unfavorable in St. Petersburg. In addition, it is necessary to further consider other aspects of micromobility, such as problems concerning equity, which have so far not been accounted for [79,80].

**Author Contributions:** Conceptualization, D.B., V.L. and I.S.; methodology, D.B. and V.L., formal analysis, D.B. and V.L.; investigation, D.B., V.L. and I.S.; data curation, V.L.; writing—original draft preparation, D.B. and V.L.; writing—review and editing, I.S.; visualization, D.B.; supervision, D.B.; funding acquisition, I.S. All authors have read and agreed to the published version of the manuscript.

**Funding:** This research received no external funding.

**Institutional Review Board Statement:** Ethical approval was received from the Ethics Commission founded in the Institute of Humanities, Peter the Great St. Petersburg Polytechnic University (26.02.21 No. 7), which is ruled by the code of ethics of the Russian Society of Sociologists.

**Informed Consent Statement:** Informed consent was obtained from all subjects involved in the study.

**Data Availability Statement:** The data presented in this study are available on request from the corresponding author. The data are not publicly available due to restrictions of privacy.

**Conflicts of Interest:** The authors declare no conflict of interest.

## Appendix A

**Table A1.** Results of a Survey of Residents of St. Petersburg Using Shared Micromobility.

| | Judgment Issues (Bike) | Mean Observation | Judgment Issues (Scooter) | Mean Observation |
|---|---|---|---|---|
| Micro-vehicle experience (riding of Micro-vehicle) | | 6.9 | | 6.68 |
| Satisfaction with user experience | | 7.15 | | 7.2 |
| | Bike-sharing is fun | 7.2 | Scooter-sharing is fun | 7.3 |
| | I believe that bike-sharing is useful | 7.3 | I believe that scooter-sharing is useful | 7.2 |
| | Bike-sharing is a pleasure | 7.3 | Scooter-sharing is a pleasure | 7.4 |
| | Bike-sharing is exciting | 6.8 | Scooter-sharing is exciting | 7.0 |
| | | 6.85 | | 6.85 |
| Trust Attitude | Based on my own experience with bike-sharing, I know it is a reliable system | 6.9 | Based on my own experience with scooters, I know it meets the expectations of the users | 6.8 |
| | Based on my own bike-sharing experience, I know this system is harmless | 6.8 | Based on my own scooter-sharing experience, I know this system is harmless | 6.9 |

**Table A1.** *Cont.*

|  | Judgment Issues (Bike) | Mean Observation | Judgment Issues (Scooter) | Mean Observation |
|---|---|---|---|---|
| Personal Accomplishment |  | 6.8 |  | 6.05 |
|  | I am glad when I see how much I have cycled | 6.8 | I am glad when I see how much I have ridden on the scooter | 6.2 |
|  | I feel energized when I see calories burned | 6.8 | I feel energized when I see calories burned | 5.9 |
| Perceived Usefulness |  | 6.9 |  | 6.9 |
|  | Bike-sharing helps me | 6.9 | Scooter-sharing helps me | 6.9 |
|  | Bike-sharing can make my journeys more comfortable | 7.1 | Scooter-sharing can make my journeys more comfortable | 7.1 |
|  | Bike-sharing can make my journeys more efficient | 7.0 | Scooter-sharing can make my journeys more efficient | 7.1 |
|  | I believe that bike-sharing is practical | 7.1 | I believe that scooter-sharing is practical | 7.0 |
|  | I believe bike-sharing is good for my daily commute | 6.5 | I believe scooter-sharing is good for my daily commute | 6.6 |
| Subjective Well-Being |  | 6.8 |  | 6.4 |
|  | My bike-sharing experience was memorable—it made my life easier | 6.7 | My scooter-sharing experience was memorable—it made my life easier | 6.6 |
|  | In general, shortly after riding the shared bike, I felt good | 7.1 | In general, shortly after riding the shared scooter, I felt good | 6.7 |
|  | After using bike sharing, I felt like my life had become more complete | 6.6 | After using scooter sharing, I felt like my life had become more complete | 5.9 |
| Social environment |  | 6 |  | 5.68 |
| Influence of public opinion |  | 6.1 |  | 5.43 |
|  | When I rent a bike, I look better in the eyes of others | 6.0 | When I rent a scooter, I look better in the eyes of others | 5.2 |
|  | Using bike sharing gives me social approval | 6.3 | Using scooter sharing gives me social approval | 5.4 |
|  | The fact that I rent a bike makes a good impression on those around me | 6.0 | The fact that I rent a scooter makes a good impression on those around me | 5.7 |
| Influence of the reference group |  | 5.86 |  | 5.56 |
|  | Most of the people who are important to me think that I should use bike-sharing | 5.8 | Most of the people who are important to me think that I should use scooter-sharing | 5.5 |
|  | Most of the people who are important to me think that cycling is a good idea | 5.8 | Most of the people who are important to me think that riding a scooter is a good idea | 5.5 |
|  | Most of the people who influence my decisions think that I should use bike sharing | 6.0 | Most of the people who influence my decisions think that I should use scooter sharing | 5.7 |

**Table A1.** *Cont.*

| | Judgment Issues (Bike) | Mean Observation | Judgment Issues (Scooter) | Mean Observation |
|---|---|---|---|---|
| **Impact on the immediate environment** | | 6.06 | | 6.06 |
| | I have often recommended bike sharing to my family and friends | 6.3 | I have often recommended scooter sharing to my family and friends | 6.2 |
| | I often use bike-sharing as a way to relax with my family and friends | 6.0 | I often use scooter-sharing as a way to relax with my family and friends | 6.3 |
| | My friends and family started using bike-sharing on my advice | 5.9 | My friends and family started using scooter-sharing on my advice | 5.7 |
| **Perception of micromobility** | | 6.77 | | 6.86 |
| **Injury risk** | | 6.2 | | 6.6 |
| | Inexperienced people using bike-sharing can get hurt | 6.0 | Inexperienced people using scooter-sharing can get hurt | 6.0 |
| | Because of the small number of cycle lanes, a cyclist can be the culprit in sidewalk accidents | 6.6 | Because of the small number of scooter lanes, a scooter-user can be the culprit in sidewalk accidents | 7.0 |
| | Inexperienced bike-sharing users can injure others | 6.0 | Inexperienced scooter-sharing users can injure others | 6.8 |
| **Environmental friendliness** | | 6.86 | | 6.56 |
| | I use bike-sharing because a bike is an environmentally friendly mode of transport | 6.8 | I use scooter-sharing because a scooter is an environmentally friendly mode of transport | 6.5 |
| | The more people use bike-sharing, the less we pollute the environment | 7.1 | The more people use scooter-sharing, the less we pollute the environment | 6.8 |
| | I advise my friends and family to use bike-sharing whenever possible, as bicycles are environmentally friendly | 6.7 | I advise my friends and family to use a scooter-sharing whenever possible, as the scooter is environmentally friendly | 6.4 |
| **Convenience of the digital environment** | | 7.0 | | 7.06 |
| | I like that bicycles are equipped with GPS modules | 7.0 | I like that scooters are equipped with GPS modules | 6.7 |
| | I like that I can see the nearest bike-sharing station in the mobile app | 7.3 | I like that I can see the nearest scooter-sharing station in the mobile app | 7.5 |
| | I never had a problem getting my bike to the station | 6.7 | I never had a problem getting my scooter to the station | 7.0 |
| **Perceived Ease of Use** | | 7.03 | | 7.23 |
| | I can easily and simply interact with the bike sharing information system | 7 | I can easily and simply interact with the scooter sharing information system | 7.3 |
| | Bike-sharing is easy for me | 7.0 | Scooter-sharing is easy for me | 7.2 |
| | I can easily and simply interact with bike sharing services | 7.1 | I can easily and simply interact with scooter sharing services | 7.2 |

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
