# Peer review of "Shared Micromobility: Between Physical and Digital Reality"

_sustainability, doi:10.3390/su14042467_

Round 1
Reviewer 1 Report
The manuscript focuses on a topical issue.
The manuscript still has some grammatical and formatting errors. It is considered appropriate to specify the reason for some words with capital letters or to insert acronyms.
Part of the figure in Figure 1 needs to be corrected where it says "environmental costs".
In figure 2 it is good to move the word "USEFULNESS".
Author Response
Dear reviewers,
Thank you very much for the recommendations and positive evaluation of our work!
Reviewer 1
The manuscript focuses on a topical issue.
The manuscript still has some grammatical and formatting errors. It is considered appropriate to specify the reason for some words with capital letters or to insert acronyms.
Part of the figure in Figure 1 needs to be corrected where it says "environmental costs".
In figure 2 it is good to move the word "USEFULNESS".
Thank you very much for your valuable comments, we edited the figures and removed capital letters in the indicator titles.
Reviewer 2
The changes in the article give a great picture of its reference and make it more detailed. However, I am concerned about the research effort as it is relatively small and 100% of the people are students. How does this relate to the mapping of the behavior of city residents?
We insert a paragraph to methodology section that shows the features of the early stage of development of micromobility in St. Petersburg. As well as a link to an article indicating that scooters in St. Petersburg are mainly used by students
Reviewer 3
All my comments have been addressed. Congratulations for the good work.
Thank you!
Reviewer 2 Report
The changes in the article give a great picture of its reference and make it more detailed. However, I am concerned about the research effort as it is relatively small and 100% of the people are students. How does this relate to the mapping of the behavior of city residents?
Author Response

(The authors gave the same response as above.)

Reviewer 3 Report
All my comments have been addressed. Congratulations for the good work.
Author Response
Dear reviewers,
Thank you very much for the recommendations and positive evaluation of our work!
Reviewer 1
The manuscript focuses on a topical issue.
The manuscript still has some grammatical and formatting errors. It is considered appropriate to specify the reason for some words with capital letters or to insert acronyms.
Part of the figure in Figure 1 needs to be corrected where it says "environmental costs".
In figure 2 it is good to move the word "USEFULNESS".
Thank you very much for your valuable comments, we edited the figures and removed capital letters in the indicator titles.
Reviewer 2
The changes in the article give a great picture of its reference and make it more detailed. However, I am concerned about the research effort as it is relatively small and 100% of the people are students. How does this relate to the mapping of the behavior of city residents?
We insert a paragraph to methodology section that shows the features of the early stage of development of micromobility in St. Petersburg. As well as a link to an article indicating that scooters in St. Petersburg are mainly used by students
Reviewer 3
All my comments have been addressed. Congratulations for the good work.
Thank you!
This manuscript is a resubmission of an earlier submission. The following is a list of the peer review reports and author responses from that submission.
Round 1
Reviewer 1 Report
The very idea of conducting the research is very positive, after the indicated questions, I think that one could have much more from this material. The analysis at the level of numerical or percentage shares is really low. There is no reference to sustainable development. I am talking about static analysis but there are no calculations in this regard.
Author Response
1st review
The very idea of conducting the research is very positive, after the indicated questions, I think that one could have much more from this material. The analysis at the level of numerical or percentage shares is really low. There is no reference to sustainable development. I am talking about static analysis but there are no calculations in this regard.
Thank you very much for your positive feedback. We have broadened the description of the results and added sustainability references.
We thought about using this statistical analysis but decided not to use it in this paper, as it is already long and overloaded. Thanks!
Reviewer 2 Report
Not sure what's the utility of this paper at all other than indicating the obvious observations. Furthermore, survey questions do not sound all the useful. For example, The questionnaire statement in line 381-382 and page 12 don’t sound even relevant…
Author Response
2nd review
Not sure what's the utility of this paper at all other than indicating the obvious observations. Furthermore, survey questions do not sound all the useful. For example, The questionnaire statement in line 381-382 and page 12 don’t sound even relevant
Thank you for your valuable comments. We try to highlight the scientific novelty of the research.
We agree that some statements look a little bit pretentious, especially when translated into English. But this was done for a clearer distinction between the analyzed aspects. In our new version of research, we made a clearer translation.
In addition, We would like to note that the close in meaning statements were used for evaluation in other researches, for example Ma, L., Zhang, X., Ding, X., & Wang, G., (2018), Bike sharing and users’ subjective well-being: An empirical study in China, Transportation Research Part A: Policy and Practice, vol. 118, Cham: Springer pp. 14–24. https://doi.org/10.1016/j.tra.2018.08.040
Reviewer 3 Report
The paper presents a social study on users' perception of shared micromobility services in the city of St. Petersburg. Also, in the introduction, the paper clearly presents the challenges that this type of mobility has been facing over the last few years. Concern about the emission of polluting gases, user (and pedestrian) safety and regional policies on the subject were duly addressed. Finally, an interdisciplinary survey was carried out with city dwellers in order to define, by mathematical and statistical methods, the perception of the shared micromobility service.
In section 3.1, the authors present a history of the installation of micromobility in the city of St. Petersburg. Definitely the city, in the last years, went through an important discussion about this type of service, considering the entrance/exit of several specialized companies and the several negative impacts caused by bikes and e-scooters. Thus, a statistical analysis of service satisfaction, pointing out the advantages and disadvantages, is very importance for the evolution, adjustments and implementation of shared micromobility in the city, showing the importance of the work presented in this paper.
The survey and results compilation presented are satisfactory, meeting the need to present the perception (from a statistical point of view) of users of shared micromobility.
In the discussion section, the authors managed to compare the profile of the city of St. Petersburg with statistical data of other cities profiles, proving that this type of service is usual and common in cities around the world. Furthermore, the authors obtain, based on the data presented, conclusions about the use/perception of the shared micromobility service in St. Petersburg, which is very desirable for this type of study.
Below I provide some suggestions for improving the text:
Fix the typo "Nevertheless, , " in line 43.
Improve the quality (placement of subtitles) in Figure 2.
Fix the typo "usedscooters, , " in line 223.
correct the typo "km2" in line 251
correct the typo "in 2018 - 56 stations" in line 257
The authors quote in line 262 "the company was forced to install new primitive bicycle-parking lots". I suggest insert a image of this primitive parking lots to contrast the image provided in Figure 3.
correct the typo "12500scooters" in line 286
correct the typo "than 250 W). - from 2500-20000" in line 297
correct the typo "public outcry.. An" in line 317
correct the typo "friends,’. , Conversely" in line 394
correct the typo "is a good idea’ (5.5)." in line 402
When commenting on the results in Figures 8-11 the authors use different "patterns" throughout the text. Time uses the terms enclosed in quotation marks ("), time uses a dash for (-) to present the numerical result, etc. I suggest standardizing the text.
correct the typo "‘just" in line 446
Author Response
3rd review
The paper presents a social study on users' perception of shared micromobility services in the city of St. Petersburg. Also, in the introduction, the paper clearly presents the challenges that this type of mobility has been facing over the last few years. Concern about the emission of polluting gases, user (and pedestrian) safety and regional policies on the subject were duly addressed. Finally, an interdisciplinary survey was carried out with city dwellers in order to define, by mathematical and statistical methods, the perception of the shared micromobility service.
In section 3.1, the authors present a history of the installation of micromobility in the city of St. Petersburg. Definitely the city, in the last years, went through an important discussion about this type of service, considering the entrance/exit of several specialized companies and the several negative impacts caused by bikes and e-scooters. Thus, a statistical analysis of service satisfaction, pointing out the advantages and disadvantages, is very importance for the evolution, adjustments and implementation of shared micromobility in the city, showing the importance of the work presented in this paper.
The survey and results compilation presented are satisfactory, meeting the need to present the perception (from a statistical point of view) of users of shared micromobility.
In the discussion section, the authors managed to compare the profile of the city of St. Petersburg with statistical data of other cities profiles, proving that this type of service is usual and common in cities around the world. Furthermore, the authors obtain, based on the data presented, conclusions about the use/perception of the shared micromobility service in St. Petersburg, which is very desirable for this type of study.
Thank you for carefully reading the work and your comments.
Below I provide some suggestions for improving the text:
Fix the typo "Nevertheless, , " in line 43.
Thanks. Done.
Improve the quality (placement of subtitles) in Figure 2.
Thanks. Done.
Fix the typo "usedscooters, , " in line 223.
Thanks. Done.
correct the typo "km2" in line 251
Thanks. Done.
correct the typo "in 2018 - 56 stations" in line 257
Thanks. Done.
The authors quote in line 262 "the company was forced to install new primitive bicycle-parking lots". I suggest insert a image of this primitive parking lots to contrast the image provided in Figure 3.
Thanks. Done.
correct the typo "12500scooters" in line 286
Thanks. Done.
correct the typo "than 250 W). - from 2500-20000" in line 297
Thanks. Done.
correct the typo "public outcry.. An" in line 317
Thanks. Done.
correct the typo "friends,’. , Conversely" in line 394
Thanks. Done.
correct the typo "is a good idea’ (5.5)." in line 402
Thanks. Done.
When commenting on the results in Figures 8-11 the authors use different "patterns" throughout the text.
Thanks. Done.
Time uses the terms enclosed in quotation marks ("), time uses a dash for (-) to present the numerical result, etc. I suggest standardizing the text.
Thanks. Done.
correct the typo "‘just" in line 446
Thanks. Done.
Reviewer 4 Report
The manuscript focuses on a topic of current importance.
It is considered appropriate to include more bibliographical references relating to micro-mobility and its possession or sharing, highlighting the spread of this type of mobility in different parts of the world.
1) Raptopoulou, A., Basbas, S., Stamatiadis, N., & Nikiforiadis, A. (2020, June). A first look at e-scooter users. In Conference on Sustainable Urban Mobility (pp. 882-891). Springer, Cham.
2)Hosseinzadeh, A., Algomaiah, M., Kluger, R., & Li, Z. (2021). E-scooters and sustainability: Investigating the relationship between the density of E-scooter trips and characteristics of sustainable urban development. Sustainable cities and society, 66, 102624.
3) Campisi, T., Nahiduzzaman, K. M., Ticali, D., & Tesoriere, G. (2020, July). Bivariate analysis of the influencing factors of the upcoming personal mobility vehicles (PMVs) in Palermo. In International Conference on Computational Science and Its Applications (pp. 868-881). Springer, Cham.
It is appropriate to highlight in the introduction the benefits and criticalities of the spread of micro-mobility by referring to the concept of equity and accessibility of cities as well as sustainability and resilience.
We therefore recommend reading the following works
1) Johnston, K., Oakley, D., Durham, A. V., Bass, C., & Kershner, S. (2020). Regulating Micromobility: Examining Transportation Equity and Access. JCULP, 4, 682.
2) Stowell, H. G. (2020). Making micromobility equitable for all. Institute of Transportation Engineers. ITE Journal, 90(2), 46-49.
In the light of the recent pandemic, it is necessary to point out whether this has affected modal choices and in what way
More explanation is needed to accompany figures 1 and 4.
In figure 2, there seems to be some missing parts.
Acronyms should be written out in full when used in the text for the first time.
The novelty of the research should be highlighted in the introduction.
In addition, it is necessary to point out the limitations of the present work in the conclusion.
Author Response
4 review
The manuscript focuses on a topic of current importance.
It is considered appropriate to include more bibliographical references relating to micro-mobility and its possession or sharing, highlighting the spread of this type of mobility in different parts of the world.
1) Raptopoulou, A., Basbas, S., Stamatiadis, N., & Nikiforiadis, A. (2020, June). A first look at e-scooter users. In Conference on Sustainable Urban Mobility (pp. 882-891). Springer, Cham.
2)Hosseinzadeh, A., Algomaiah, M., Kluger, R., & Li, Z. (2021). E-scooters and sustainability: Investigating the relationship between the density of E-scooter trips and characteristics of sustainable urban development. Sustainable cities and society, 66, 102624.
3) Campisi, T., Nahiduzzaman, K. M., Ticali, D., & Tesoriere, G. (2020, July). Bivariate analysis of the influencing factors of the upcoming personal mobility vehicles (PMVs) in Palermo. In International Conference on Computational Science and Its Applications (pp. 868-881). Springer, Cham.
It is appropriate to highlight in the introduction the benefits and criticalities of the spread of micro-mobility by referring to the concept of equity and accessibility of cities as well as sustainability and resilience.
We therefore recommend reading the following works
1) Johnston, K., Oakley, D., Durham, A. V., Bass, C., & Kershner, S. (2020). Regulating Micromobility: Examining Transportation Equity and Access. JCULP, 4, 682.
2) Stowell, H. G. (2020). Making micromobility equitable for all. Institute of Transportation Engineers. ITE Journal, 90(2), 46-49.
Thank you very much for attentive reviewing the work and valuable recommendation. We added recommended resources.
Here are overview of micromodity in different countries
Micromobility as a new andsignificant phenomenon of urban life in recent years has attracted the attention of researchers around the world, specifically in: Greece [20], the USA [21], Italy [22], Germany [23], Denmark [24], Thailand [12], China [25], Austria [26], France [27], Australia [28], New Zealand [29], Saudi Arabia [30], and Singapore [31].
Here is a more detailed description of the connection of the topic with sustainability
Hosseinzadeh, Algomaiah, Kluger, Li consider micromobility as part of the Smart City paradigm [15] which, according to many authors, is the key to enhancing sustainable urban development [16]. Information and communication technology is becoming an essential element for accelerating progress on achieving sustainability [17]. Sustainable transportation in cities should strive to optimize their environmental impact (e.g., reducing greenhouse gas emissions, distance, fuel consumption, , pollution, empty miles whilst increasing vehicle load) and social impact (e.g., increasing accessibility, reliability, health and safety whilst reducing congestion) [18]. This study offers a multilevel consideration of Cyber -physical-social reality. Studying this phenomenon, taking into account the digital, social and physical dimensions, and reveals weaknesses and potential in the development of two-wheeled vehicles as a way of contributing to sustainable urban development.
Since the topic of equality, unfortunately, remained outside the scope of this work, we noted this in the limits of the study at discussion.
This study is limited to a specific city case at a fairly early stage in the use of shared micromobility. In addition to the factors considered, there are other factors affecting the use of shared micromobility services, in particular weather conditions [76], which are also often unfavorable in St. Petersburg. In addition, it is necessary to further consider aspects of micromobility, such as problems concerning equity, which have so far not been accounted for. [77,78]
In the light of the recent pandemic, it is necessary to point out whether this has affected modal choices and in what way
Thank you! We highlighted what effect could be and that there is no such big influence in St Petersburg because of initial stage of development (in discussion part)
In cities where micro-exchange is a well-known reality, attitudes towards it have changed in a positive direction due to the need to keep distance during the Covid-19 pandemic [67]. In St. Petersburg, micromobility sharing has become a novelty in recent years. It should be noted that the launch of micromobility during a pandemic placed it in a uniquely advantageous position. This said, we do not have the opportunity to compare its use with the period before isolation
More explanation is needed to accompany figures 1 and 4.
Thanks. Done.
figure 1
Considering the relationship between the user and the digital environment, shared micromobility represents the ideal of public transport, when transport as an object of the physical environment is ideally integrated into the digital-physical system (Fig. 1) of round-the-clock satisfaction of the urgent need of citizens for movement. The digital environment of micromobility represents a reflection of the key factors of the physical environment, primarily demonstrating the location of the nearest vehicles and possible parking spots. In addition, the digital environment permits an to optimization of the route by means of maps which reflect the situation on the road. This said, many important factors of the physical environment today are still not reflected in cyber reality.
figure 4
Fig. 4 shows the frequency of use of scooters. 52% indicated the answer ‘from one to several times a month’, 22% of respondents use the service ‘from one to several times a week’ and 26% of them use it ‘from one to several times a year’. This indicates that the majority of respondents quite often use micromobility services in their daily life, and that for them it is a permanent mode of transport.
In figure 2, there seems to be some missing parts.
Sorry for the confusion, the picture was improved, circles represent other people, not ideas
Acronyms should be written out in full when used in the text for the first time.
Thanks. Done.
The novelty of the research should be highlighted in the introduction.
Thanks. Done.
In addition, it is necessary to point out the limitations of the present work in the conclusion.
Thanks. Done.
Round 2
Reviewer 1 Report
The article still lacks a summary, the authors only indicated the discussion. There is no clearly defined research model or hypotheses, which distorts the evaluation process. Because the assumptions are not fully known
Author Response
Dear reviewers, thank you for your valuable comments.
Thanks for editing the article by specialist of the journal.
We tried to take into account all the comments, as well as correct the style and make the text clearer.
Reviewer 1
The article still lacks a summary, the authors only indicated the discussion. There is no clearly defined research model or hypotheses, which distorts the evaluation process. Because the assumptions are not fully known
Thanks for your review. We have added research objectives to the introduction and revised the summarizing part of the discussion. All those are highlighted in blue. We did not make a summary separately, since according to the recommendations it is applied "if the discussion is unusually long or complex"
Reviewer 4
The manuscript still has some grammatical errors.
thanks, fixed
Perhaps a small legend accompanying figures 8 to 11 with the different levels of judgement would help in understanding the results obtained.
Thanks you. Done
Reviewer 4 Report
The manuscript still has some grammatical errors.
Perhaps a small legend accompanying figures 8 to 11 with the different levels of judgement would help in understanding the results obtained.
Author Response

(The authors gave the same response as above.)

Round 3
Reviewer 1 Report
The article has a lot of editorial errors, but the changes introduced already give a picture of the study. There is still no definition of the aim or research question and hypotheses, but the research itself is described clearly and lucidly.